# AutoNeRF: Training Implicit Scene Representations with Autonomous Agents

## Abstract

Implicit representations such as Neural Radiance Fields (NeRF) have been shown to be very effective at novel view synthesis. However, these models typically require manual and careful human data collection for training. In this paper, we present AutoNeRF, a method to collect data required to train NeRFs using autonomous embodied agents. Our method allows an agent to explore an unseen environment efficiently and use the experience to build an implicit map representation autonomously. We compare the impact of different exploration strategies including handcrafted frontier-based exploration, end-to-end and modular approaches composed of trained high-level planners and classical low-level path followers. We train these models with different reward functions tailored to this problem and evaluate the quality of the learned representations on four different downstream tasks: classical viewpoint rendering, map reconstruction, planning, and pose refinement. Empirical results show that NeRFs can be trained on actively collected data using just a single episode of experience in an unseen environment, and can be used for several downstream robotic tasks, and that modular trained exploration models outperform other classical and end-to-end baselines. Finally, we show that AutoNeRF can reconstruct large-scale scenes, and is thus a useful tool to perform scene-specific adaptation as the produced 3D environment models can be loaded into a simulator to fine-tune a policy of interest.

## 1 Introduction

Exploration is a key challenge in building autonomous navigation agents that operate in unseen environments. In the last few years, there has been a significant amount of work on training exploration policies to maximize coverage (Chaplot et al., 2020b; Chen et al., 2019; Savinov et al., 2018), find goals specified by object categories (Gupta et al., 2017; Chaplot et al., 2020a; Marza et al., 2022; Ramakrishnan et al., 2022; Ramrakhya et al., 2022), images (Zhu et al., 2017; Chaplot et al., 2020d; Hahn et al., 2021; Mezghan et al., 2022) or language (Anderson et al., 2018b; Krantz et al., 2020; Min et al., 2022) and for embodied active learning (Chaplot et al., 2020c; 2021). Among these methods, modular learning methods have shown to be very effective at various embodied tasks (Chaplot et al., 2020b;a; Deitke et al., 2022; Gervet et al., 2022). These methods learn an exploration policy that can build an explicit semantic map of the environment which is then used for planning and downstream embodied AI tasks such as Object Goal or Image Goal Navigation.

Concurrently, in the computer graphics and vision communities, there has been a recent but large body of work on learning implicit map representations, particularly based on Neural Radiance Fields (NeRF) (Mildenhall et al., 2020; Müller et al., 2022; Garbin et al., 2021; Yu et al., 2021; Xie et al., 2021). Prior methods (Tancik et al., 2022; Vora et al., 2021; Zhi et al., 2021a;b) demonstrate strong performance in novel view synthesis and are appealing from a scene understanding point of view as a compact and continuous representation of appearance and semantics in a 3D scene. However, most approaches building implicit representations require data collected by humans (Mildenhall et al., 2020; Tancik et al., 2022; Zhi et al., 2021b). Can we train embodied agents to explore an unseen environment efficiently to collect data that can be used to create implicit map representations or NeRFs autonomously? In this paper, our objective is to tackle this problem of active exploration for autonomous NeRF construction. If an embodied agent is able to build an implicit map representation autonomously, it can then use it for a variety of downstream tasks such as planning, pose estimation, and navigation. Just a single episode or a few minutes of exploration in an unseen environment can

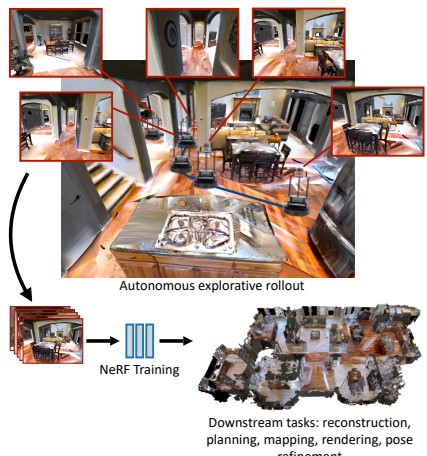

Figure 1: We propose a method to automatically generate 3D models of a scene by training NeRFs on data collected by autonomous agents. We compare policies (classical and RL-trained with different reward functions) by evaluating the quality of the final NeRF on reconstruction, planning, mapping, rendering, and pose refinement.

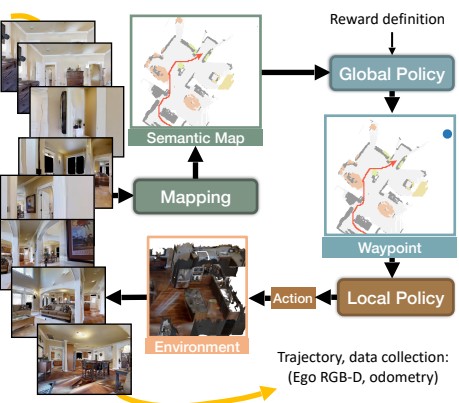

Figure 2: We adapt the modular policy in Chaplot et al. (2020a): a mapping module generates a semantic and occupancy top-down map from ego-centric RGB-D observations and sensor pose. A high-level policy trained with RL predicts global waypoints, which are followed by a low-level policy (fast marching). The sequence of observations comprises the data input to NeRF training.

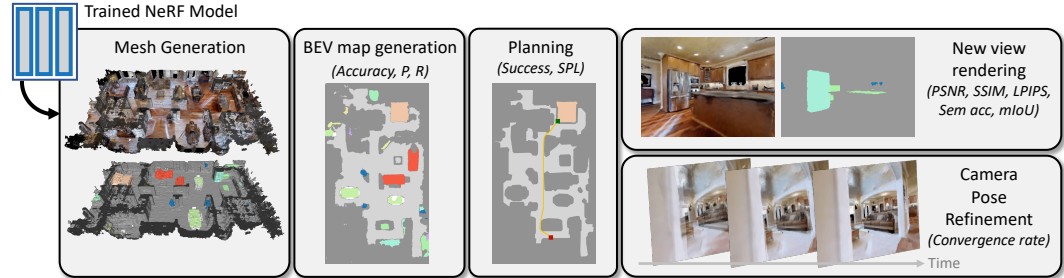

Figure 3: **Downstream tasks** — the model trained from autonomously collected data is used for several downstream tasks related to robotics: Mesh generation for the covered scene (color or semantic mesh); Birds-eye-view map generation and navigation/planning on this map; new view generation of RGB and semantic frames; camera pose refinement (visual servoing).

be sufficient to build an implicit representation that can be utilized for improving the performance of the agent in that environment for several tasks without any additional supervision.

In this work, we introduce AutoNeRF, a modular policy trained with Reinforcement Learning (RL) that can explore an unseen 3D scene to collect data for training a NeRF model autonomously (Figure 1). While most prior work evaluates NeRFs on rendering quality, we propose a range of downstream tasks to evaluate them (and indirectly, the exploration policies used to gather data for training these representations) for Embodied AI applications. Specifically, we use geometric and semantic map prediction accuracy, planning accuracy for Object Goal and Point Goal navigation and camera pose refinement (Figure 3). We show that AutoNeRF outperforms the well-known frontier exploration algorithm as well as state-of-the-art end-to-end learnt policies, and also study the impact of different reward functions on the downstream performance of the NeRF model. We also study how AutoNeRF can be used as a tool to autonomously adapt policies to a specific scene at deployment time by providing a high-quality reconstruction of large-scale environments that can be loaded into a simulator to improve the performance of any given agent safely.

## 2 RELATED WORK

**Neural fields —** represent the structure of a 3D scene with a neural network. They were initially introduced in Mescheder et al. (2019); Park et al. (2019); Chen & Zhang (2019) as an alternative to discrete representations such as voxels (Maturana & Scherer, 2015), point clouds (Fan et al., 2017) or meshes (Groueix et al., 2018). Neural Radiance Fields (NeRF) (Mildenhall et al., 2020) then

introduced a differentiable volume rendering loss allowing to supervise 3D scene reconstruction from 2D supervision, achieving state-of-the-art performance on novel view synthesis. Follow-up work has addressed faster training and inference (Müller et al., 2022; Garbin et al., 2021), or training from few images (Yu et al., 2021). Xie et al. (2021) references advances in this growing field.

**Neural fields in robotics —** implicit representations have also been proposed for real-time SLAM (Sucar et al., 2021; Zhu et al., 2022; 2023). Zhu et al. (2022) introduced a hierarchical implicit representation to represent large scenes and Zhu et al. (2023) performed SLAM without requiring depth information. Zhi et al. (2021a) augmented NeRFs with a semantic head trained from sparse and noisy 2D semantic maps. Implicit representations can map occupancy, explored area, and semantic objects to navigate towards (Marza et al., 2023), or the density of a scene for drone obstacle avoidance (Adamkiewicz et al., 2022). They have also been used for camera pose refinement through SGD directly on a loss in rendered pixel space (Yen-Chen et al., 2021). In contrast to the literature, we investigate training these representations from data collected by autonomous agents directly and explore the effect of the choice of policy on downstream robotics tasks.

**Active learning for neural fields —** has not yet been extensively studied. Most works focus on fixed datasets of 2D frames and tackle the active selection of training data. ActiveNeRF (Pan et al., 2022) estimates the uncertainty of a NeRF model by expressing radiance values as Gaussian distributions. ActiveRMAP (Zhan et al., 2022) minimizes collisions and maximizes an entropy-based information gain metric. These methods target rather small scenes in non-robotic scenarios, either single objects or forward-facing only. In contrast, we start from unknown environments and actively explore large indoor scenes requiring robotic exploration policies capable of handling complex scene understanding and navigation.

**Autonomous scene exploration —** is generally defined as a coverage maximization problem, a baseline being Frontier Based Exploration (FBE) (Yamauchi, 1997). Different variants exist (Dornhege & Kleiner, 2013; Holz et al., 2010; Xu et al., 2017) but the core principle is to maintain a frontier between explored and unexplored space and to sample points on it. Learning-based approaches are explored in recent work (Chaplot et al., 2020b; Chen et al., 2019; Savinov et al., 2018; Ramakrishnan et al., 2021). In this work, we study how different definitions of exploration impact the quality of an implicit scene representation.

## 3 BACKGROUND

### 3.1 MODULAR EXPLORATION POLICIES

The trained policy aims to explore a 3D scene to collect a sequence of 2D RGB and semantic frames as well as camera poses, that will be used to train a NeRF model. Following Chaplot et al. (2020a;b), we adapt a modular policy composed of a *Mapping* process that builds a semantic map, a *Global Policy* that outputs a global waypoint from the semantic map as input, and finally, a *Local Policy* that navigates towards the global goal, see Figure 2.

**Semantic Map —** a 2D top-down map is maintained at each time step $t$, with several components: (i) an occupancy component $\mathbf{m}_t^{occ} \in \mathbb{R}^{M \times M}$ stores information on free navigable space; (ii) an exploration component $\mathbf{m}_t^{exp} \in \mathbb{R}^{M \times M}$ sets to 1 all cells which have been within the agent's field of view since the beginning of the episode; (iii) a semantic component $\mathbf{m}_t^{sem} \in \mathbb{R}^{S \times M \times M}$, where $M \times M$ is the spatial size and $S$ denotes the number of channels storing information about the scene. Additional maps store the current and previous agent locations. Egocentric maps are updated by inverse projection from the depth frames and pooling to the ground plane, and are integrated over time taking into account agent poses estimated from sensor information. The semantic maps additionally use predictions obtained with Mask R-CNN (He et al., 2017).

**Policies —** intermediate waypoints are predicted by the *Global Policy*, a convolutional neural network taking as input the stacked maps (we follow Chaplot et al. (2020a)) and is trained with RL/PPO (Schulman et al., 2017). A *Local Policy* navigates towards the waypoint taking discrete actions for 25 steps following the path planned using the *Fast Marching Method* (Sethian, 1996).

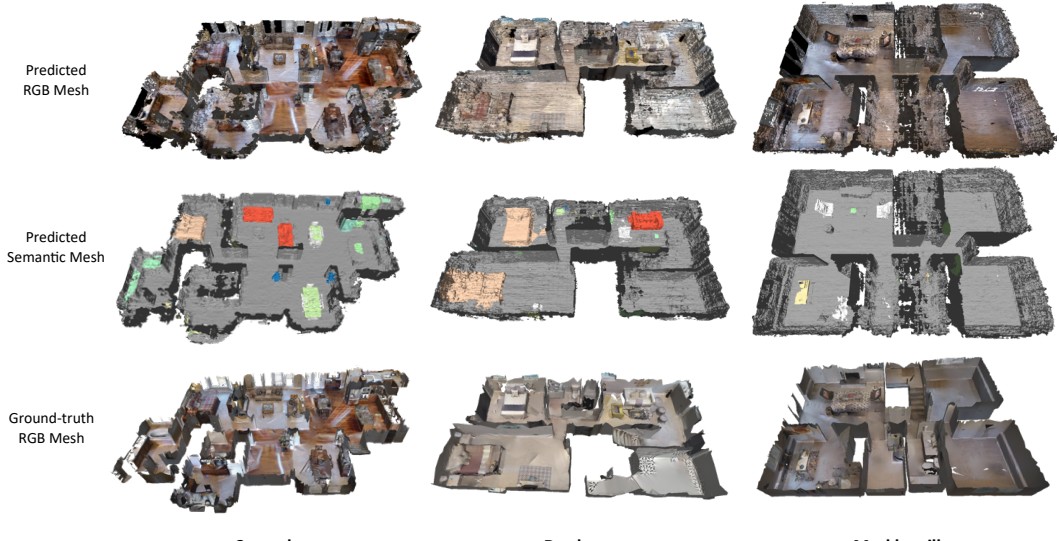

**Figure 4: Mesh reconstruction**: reconstruction of 3 Gibson val scenes extracted from a NeRF model trained on data gathered by our modular policy (*Ours (obs.)*). Both geometry, semantics, and appearance are satisfying.

## 3.2 NEURAL RADIANCE FIELDS

**Vanilla Semantic NeRF —** Neural Radiance Fields (Mildenhall et al., 2020) are composed of MLPs predicting the density $\sigma$, color $c$ and, eventually as in Zhi et al. (2021a), the semantic class $s$ of a particular $3D$ position in space $\mathbf{x} \in \mathbb{R}^3$, given a 2D camera viewing direction $\phi \in \mathbb{R}^2$. NeRFs have been designed to render new views of a scene provided a camera position and viewing direction. The color of a pixel is computed by performing an approximation of volumetric rendering, sampling $N$ quadrature points along the ray. Given multiple images of a scene along with associated camera poses, a NeRF is trained with Stochastic Gradient Descent minimizing the difference between rendered and ground-truth images.

**Semantic Nerfacto —** we leverage recent advances to train NeRF models faster while maintaining high rendering quality and follow what is done in the Nerfacto model from Tancik et al. (2023), that we augment with a semantic head. The inputs $\mathbf{x}$ and $\phi$ are augmented with a learned appearance embedding $\mathbf{e} \in \mathbb{R}^{32}$. Both $\mathbf{x}$ and $\phi$ are first encoded using respectively a hash encoding function $h$ as $\tilde{\mathbf{x}} = h(\mathbf{x})$ and a spherical harmonics encoding function $sh$ as $\tilde{\phi} = sh(\phi)$. $\tilde{\mathbf{x}}$ is fed to an MLP $f_d$ predicting the density at the given 3D position, yielding $(\sigma, \mathbf{h_d}) = f_d(\tilde{\mathbf{x}}; \Theta_d)$, where $\mathbf{h_d}$ is a latent representation. $\mathbf{h_d}$ is fed to another MLP model $f_s$ that outputs a softmax distribution over the $S$ considered semantic classes as $\mathbf{s} = f_s(\mathbf{h_d}; \Theta_s)$ where $\mathbf{s} \in \mathbb{R}^S$. Finally, $\mathbf{h_d}$, $\tilde{\phi}$ and $\mathbf{e}$ are the inputs to $f_c$ that predicts the RGB value at the given 3D location, $\mathbf{c} = f_c(\mathbf{h_d}, \tilde{\phi}, \mathbf{e}; \Theta_c)$ where $\mathbf{c} \in \mathbb{R}^3$.

## 4 AUTONERF

We present AutoNeRF, a method to collect NeRF training data with an autonomous embodied agent. The latter is initialized in an unseen environment and must gather data in a single episode with a fixed time budget. Collected observations are then used to train a neural implicit representation of the scene (density, RGB, semantics) which is finally evaluated on several robotics-related downstream tasks: new view rendering, mapping, planning and pose refinement.

**Task Specification —** The agent is initialized at a random location in an unknown scene and at each timestep $t$ can execute a discrete action in the space $\Lambda = \{$FORWARD 25cm, TURN_LEFT 30°, TURN_RIGHT 30°$\}$. At each step, the agent receives an observation $\mathbf{o}_t$ composed of an egocentric RGB frame and a depth map. The field of view of the agent is $90°$. It also has access to odometry information. The agent can navigate for a limited number of $1500$ discrete steps.

AutoNeRF can be broken down into two phases: Exploration Policy Training and NeRF Training. In the first phase, we train an exploration policy to collect observations on a set of training scenes in a self-supervised manner, i.e. using intrinsic rewards. In the second phase, we use the trained

exploration policy to collect data in unseen test scenes, one trajectory per scene, and train a NeRF model using this data. The trained NeRF model is then evaluated on the set of downstream tasks.

## 4.1 EXPLORATION POLICY TRAINING

As described in Section 3.1, we use a modular exploration policy architecture with the *Global Policy* primarily responsible for exploration. We consider different reward signals for training the *Global Policy* tailored to our task of scene reconstruction, and which differ in the importance they give to different aspects of the scene. All these signals are computed in a self-supervised fashion using the metric map representations built by the exploration policy.

**Explored area** — (*Ours (cov.)*) optimizes the coverage of the scene, i.e. the size of the explored area, and has been proposed in the literature, e.g. in Chaplot et al. (2020a;b). It accumulates differences in the exploration component $\mathbf{m}_t^{exp}$,

$$r_t^{cov} = \sum_{i=0}^{M-1} \sum_{j=0}^{M-1} \mathbf{m}_t^{exp}[i,j] - \mathbf{m}_{t-1}^{exp}[i,j]$$

**Obstacle coverage** — (*Ours (obs.)*) optimizes the coverage of obstacles in the scene, and accumulates differences in the corresponding component $\mathbf{m}_{t-1}^{occ}[i,j]$. It targets tasks where obstacles are considered more important than navigable floor space, which is arguably the case when viewing is less important than navigating.

$$r_t^{obs} = \sum_{i=0}^{M-1} \sum_{j=0}^{M-1} \mathbf{m}_t^{occ}[i,j] - \mathbf{m}_{t-1}^{occ}[i,j]$$

**Semantic object coverage** — (*Ours (sem.)*) optimizes the coverage of the $S$ semantic classes detected and segmented in the semantic metric map $\mathbf{m}_t^{sem}$. This reward removes obstacles that are not explicitly identified as a notable semantic class — see section 5 for their definition.

$$r_t^{sem} = \sum_{i=0}^{M-1} \sum_{j=0}^{M-1} \sum_{k=0}^{S-1} \mathbf{m}_t^{sem}[i,j,k] - \mathbf{m}_{t-1}^{sem}[i,j,k]$$

**Viewpoints coverage** — (*Ours (view.)*) optimizes for the usage of the trained implicit representation as a dense and continuous representation of the scene usable to render arbitrary new viewpoints, either for later visualization as its own downstream task or for training new agents in simulation. To this end, we propose to maximize coverage not only in terms of agent positions but also in terms of agent viewpoints. Compared to Chaplot et al. (2020a), we introduce an additional 3D map $\mathbf{m}^{view}[i,j,k]$, where the first two dimensions correspond to spatial 2D positions in the scene and the third dimension corresponds to a floor plane angle of the given cell discretized into $V=12$ bins. A value of $\mathbf{m}_t^{view}[i,j,k] = 1$ indicates that cell $(i,j)$ has been seen by the agent from a (discretized) angle $k$. The reward maximizes its changes,

$$r_t^{view} = \sum_{i=0}^{M-1} \sum_{j=0}^{M-1} \sum_{k=0}^{V-1} \mathbf{m}_t^{view}[i,j,k] - \mathbf{m}_{t-1}^{view}[i,j,k]$$

## 4.2 NERF TRAINING

The sequence of observations collected by the agent comprises egocentric RGB frames $\{\mathbf{o}_t\}_{t=1...T}$, first-person semantic segmentations $\{\mathbf{s}_t\}_{t=1...T}$ and associated poses $\{\mathbf{p}_t\}_{t=1...T}$ in a reference frame, which we define as the starting position $t=0$ of each episode. In our experiments, we leverage privileged pose and semantics information from simulation. We also conduct an experiment showcasing the difference between using GT semantics from a simulator and a Mask R-CNN (He et al., 2017) model.

An important property of this procedure is that no depth information is required for reconstruction. The implicit representation is trained by mapping pixel coordinates $\mathbf{x}_i$ for each pixel $i$ to RGB values and semantic values with the volume rendering loss described in Section 3.2. The input coordinates $\mathbf{x}_i$ are obtained using the global poses $\mathbf{p}_t$ and intrinsics from calibrated cameras.

## 4.3 DOWNSTREAM TASKS

Prior work on implicit representations generally focused on two different settings: (i) evaluating the quality of a neural field based on its new view rendering abilities given a dataset of (carefully

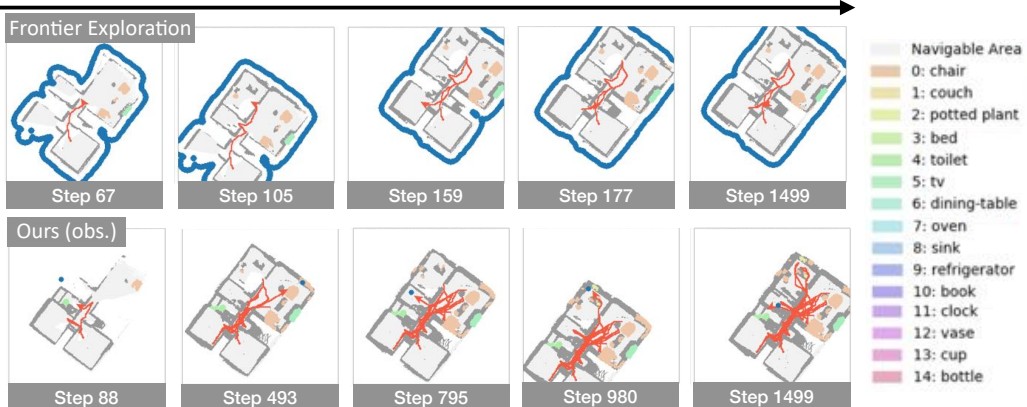

**Figure 5: Rollouts by Frontier Based exploration vs. Modular policy (*Ours (obs.)*)**: FBE properly covers the scene, but does not collect a large diversity of viewpoints, while the modular policy enters the rooms and thus provides richer training data for the neural field.

selected) training views, and (ii) evaluating the quality of a scene representation in robotics conditioned on given (constant) trajectories, evaluated as reconstruction accuracy. We cast this task in a more holistic way and more aligned with our scene understanding objective. We evaluate the impact of trajectory generation (through exploration policies) directly on the quality of the representation, which we evaluate in a goal-oriented way through multiple tasks related to robotics (cf. Figure 3).

**Task 1: Rendering —** This task is the closest to the evaluation methodology prevalent in the neural field literature. We evaluate the rendering of RGB and semantic frames as proposed in Zhi et al. (2021a). Unlike the common method of evaluating an implicit representation on a subset of frames within the trajectory, we evaluate it on a set of uniformly sampled camera poses within the scene, independently of the trajectory taken by the policy. This allows us to evaluate the representation of the complete scene and not just its interpolation ability.

We render ground-truth images and semantic masks associated with sampled camera poses using the Habitat (Savva et al., 2019; Szot et al., 2021) simulator and compare them against NeRF renderings. RGB rendering metrics are PSNR, SSIM and LPIPS (Zhang et al., 2018). Rendering of semantics is evaluated in terms of average per-class accuracy and mean intersection over union (mIoU).

**Task 2: Metric Map Estimation —** While rendering quality is linked to the perception of the scene, it is not necessarily a good indicator of its structural content, which is crucial for robotic downstream tasks. We evaluate the quality of the estimated structure by translating the continuous representation into a format, which is very widely used in map-and-plan baselines for navigation, a top-down (bird's-eye-view=BEV) map storing occupancy and semantic category information and compare it with the ground-truth from the simulator. We evaluate obstacle and semantic maps using accuracy, precision, and recall.

**Task 3: Planning —** Using maps for navigation, it is difficult to pinpoint the exact precision required for successful planning, as certain artifacts and noises might not have a strong impact on reconstruction metrics, but could lead to navigation problems. We perform goal-oriented evaluation and measure to what extent path planning can be done on the obtained top-down maps.

We sample 100 points on each scene and plan from those starting points to two types of goals: selected end positions, *PointGoal* planning, and objects categories, *ObjectGoal* planning. The latter requires planning the shortest path from the given starting point to the closest object of each semantic class available on the given scene. For both tasks, we plan with the *Fast Marching Method* and report mean *Success* and *SPL* as introduced in Anderson et al. (2018a). For a given episode, *Success* is 1 if planning stops less than 1m from from the goal, and *SPL* measures path efficiency.

**Task 4: Pose Refinement —** This task introduced in Yen-Chen et al. (2021) involves correcting an initial noisy camera pose given a RGB frame captured by the camera. We address this problem by taking the trained NeRF model, freezing its weights and optimizing the input camera pose to minimize the reconstruction difference between the NeRF-rendered frame and the provided one, as

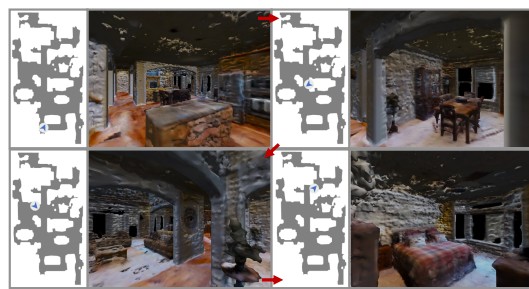

**Figure 6: Navigating in the Habitat simulator**: the underlying mesh was extracted from the trained NeRF, *Ours (cov.)*. Rendering quality and the generated BEV map are correct, as are free navigable space and collision handling. Temporal order indicated by ➡.

| Policy | Success | SPL |
|---|---|---|
| *Ft. Gibson (not comparable)*[†] | 99.7 | 97.9 |
| **Pre-trained (no ft.)** | 90.2 | 82.9 |
| **Ft. AutoNeRF** | **92.9** | **86.7** |

**Table 1: PointGoal Finetuning** – finetuning a PointGoal agent on a mesh automatically collected from a rollout and a NeRF with AutoNeRF improves mean performance over a pre-trained generic policy, which opens the door to automatic adaptions of robots to scenes after deployment. [†] an upper bound which finetunes on the original mesh. In a real use case involving a robot automatically collecting data, this mesh would not be available (not comparable).

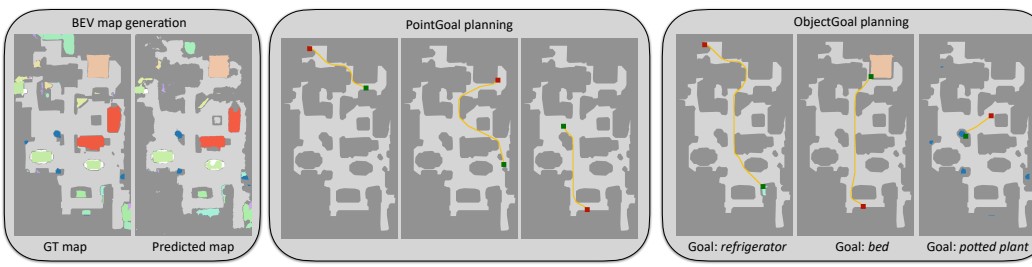

**Figure 7: BEV map tasks**: Generation of semantic BEV maps (Left), *PointGoal* (Middle) and *ObjectGoal* planning (Right).

done in Yen-Chen et al. (2021). This task is closely linked to visual servoing with a "eye-in-hand" configuration, a standard problem in robotics, in particular in its "direct" variant (Marchand, 2020), where the optimization is directly performed over losses on the observed pixel space.

To generate episodes of starting and end positions, we take $100$ sampled camera poses in each scene and apply a random transformation to generate noisy poses. The model is evaluated in terms of rotation and translation convergence rate, i.e. percentage of samples where the final difference with ground truth is less than $3°$ in rotation and $2cm$ in translation. We also report the mean translation and rotation errors for the converged samples.

## 5 EXPERIMENTAL RESULTS

**Modular Policy training —** is performed on one V100 GPU for 7 days. All modular policies are trained on the 25 scenes of the Gibson (Xia et al., 2018)-tiny training set. The used Mask R-CNN model is pre-trained on the MS COCO dataset (Lin et al., 2014) and finetuned on Gibson train scenes. We consider $S{=}15$ semantic categories: {*chair*, *couch*, *potted plant*, *bed*, *toilet*, *tv*, *dining table*, *oven*, *sink*, *refrigerator*, *book*, *clock*, *vase*, *cup*, *bottle*}.

**External baselines —** We compare our trained modular policies against the classical frontier-based exploration algorithm (*Frontier*), as well as end-to-end policies trained with RL. More specifically, we consider $4$ end-to-end policies from Ramakrishnan et al. (2021), that all share the same architecture but were trained with different exploration-related reward functions: coverage (*E2E (cov.)*), curiosity (*E2E (cur.)*), novelty (*E2E (nov.)*), reconstruction (*E2E (rec.)*). Reward functions are presented in Ramakrishnan et al. (2021).

**Evaluation —** consists in running $5$ rollouts with different start positions in each of the $5$ Gibson-tiny val scenes for each policy, always on the first house floor. A NeRF model is then trained on each trajectory data.

**NeRF models —** In our experiments, we consider two different NeRF variants presented in Section 3.2. Most experiments are conducted with *Semantic Nerfacto*, as it provides a great trade-off between training speed and quality of representation. *Semantic Nerfacto* is built on top of the *Nerfacto* model from the nerfstudio (Tancik et al., 2023) library. We augment the model with a semantic head

| Policy | RGB | | | Semantics | |
|---|---|---|---|---|---|
| | PSNR | SSIM | LPIPS | Acc. | mIoU |
| Frontier | 19.75 | 0.743 | 0.343 | 81.4 | 65.7 |
| E2E (cov.) | 20.94 | 0.750 | 0.332 | 80.1 | 63.9 |
| E2E (cur.) | 20.60 | 0.747 | 0.338 | 78.7 | 61.9 |
| E2E (nov.) | 23.36 | 0.801 | 0.268 | 84.6 | 71.4 |
| E2E (rec.) | 23.17 | 0.797 | 0.270 | 84.1 | 70.5 |
| Ours (cov.) | 24.89 | 0.837 | 0.218 | 90.2 | 81.2 |
| Ours (sem.) | 25.34 | 0.843 | 0.207 | **91.9** | 81.8 |
| Ours (obs.) | **25.56** | **0.846** | **0.203** | 91.8 | **83.2** |
| Ours (view.) | 25.17 | 0.842 | 0.211 | 91.3 | 82.0 |

Table 2: **Rendering performance** on uniformly sampled viewpoints of the full scene after training on a single trajectory.

| Policy | Occupancy | | | Semantics | | |
|---|---|---|---|---|---|---|
| | Acc. | Prec. | Rec. | Acc | Prec. | Rec. |
| Frontier | 81.2 | 86.9 | 49.9 | 99.7 | 26.6 | 21.0 |
| E2E (cov.) | 77.1 | 86.2 | 50.4 | 99.7 | 22.1 | 16.1 |
| E2E (cur.) | 81.8 | 90.3 | 50.7 | 99.7 | 19.2 | 12.5 |
| E2E (nov.) | 83.1 | 88.7 | 61.3 | 99.7 | 25.5 | 18.3 |
| E2E (rec.) | 81.6 | 87.6 | 60.0 | 99.7 | 26.2 | 18.0 |
| Ours (cov.) | 86.8 | 89.1 | 74.7 | 99.8 | 35.1 | 27.1 |
| Ours (sem.) | 86.6 | 88.3 | 76.5 | 99.8 | 35.7 | 29.8 |
| Ours (obs.) | 86.4 | 89.4 | 76.5 | 99.8 | 36.2 | 29.8 |
| Ours (view.) | **88.1** | **90.9** | **77.0** | 99.8 | **37.4** | **30.2** |

Table 3: **Map Estimation performance** – evaluation of BEV maps estimated from NeRFs.

and implement evaluation on test camera poses independently from the collected trajectory. Only the next two subsections (5.1, 5.2) will involve training a *vanilla Semantic NeRF* model, more precisely the one introduced in Zhi et al. (2021a) that also contains a semantic head. We chose this variant for these specific experiments to illustrate the possibility of providing high-fidelity representations of complex scenes, and show that a *vanilla Semantic NeRF* model trained for a longer time (12h) leads to better-estimated geometry. Results from *Semantic Nerfacto* are still very good (see Figures 10 and 9) but we found meshes to be higher quality with a vanilla NeRF model.

## 5.1 RECONSTRUCTING HOUSE-SCALE SCENES

We illustrate the possibility of autonomously reconstructing complex large-scale environments such as apartments or houses from the continuous representations trained on data collected by agents exploring the scene using the modular policy. Figure 11 shows RGB and semantic meshes for 3 Gibson val scenes. Geometry, appearance, and semantics are satisfying. In Figure 6 we show that such meshes can be loaded into the Habitat simulator and allow proper navigation and collision computations. Both occupancy top-down map generation and RGB renderings are performed by the Habitat simulator from the generated mesh.

## 5.2 AUTONOMOUS ADAPTATION TO A NEW SCENE

A long-term goal of Embodied AI is to train general policies that can be deployed on any new scene. Even such agents will likely struggle with some specificities of a given environment, and a scene-specific adaptation thus appears as a relevant solution. We explore the usage of AutoNERF to explore an environment to build a 3D representation, which is then loaded into a simulator to safely finetune a policy of interest. More specifically, we consider a depth-only PointGoal navigation policy pre-trained on Gibson. It is finetuned on 4 Gibson val scenes, using meshes generated with AutoNeRF, before being evaluated on the original Gibson meshes. Details about episodes sampling and training hyperparameters are given in the Supplementary Material.

As shown in Table 1, scene-specific finetuning on autonomously reconstructed 3D meshes allows to improve both Success and SPL. We also compare with finetuning directly on the Gibson mesh, which provides a non-comparable soft upper bound — in a real robotics scenario, these meshes would not be accessible. This shows that performance could still be improved, but it is important to note that reaching the performance of the upper bound might be about reconstructing fine details.

## 5.3 QUANTITATIVE RESULTS

**Frontier Exploration vs Modular Policy —** as can be seen in Tables 2, 3, 4, 5, RL-trained modular policies outperform frontier exploration on all metrics. This is a somewhat surprising result, since Frontier Based Exploration generally performs satisfying visual coverage of the scene, even though it can sometimes get stuck because of map inaccuracies. This shows that vanilla visual coverage, the optimized metrics in many exploration-oriented tasks, is not a sufficient criterion to collect NeRF training data. Figure 5 illustrates this point with rollouts from FBE and a modular policy trained to maximize obstacle coverage. FBE properly covers the scene but does not necessarily cover a large diversity of viewpoints, while the modular policy provides richer training data to the NeRF.

**End-to-end Policy vs Modular Policy —** Tables 2, 3, 4, 5 also show that the modular policies outperform end-to-end RL policies on all considered metrics. Interestingly, *novelty* and *reconstruc-*

| Policy | PointGoal | | ObjectGoal | |
|---|---|---|---|---|
| | Succ. | SPL | Succ. | SPL |
| Frontier | 22.4 | 21.4 | 9.6 | 9.1 |
| E2E (cov.) | 30.0 | 29.3 | 8.9 | 8.3 |
| E2E (cur.) | 29.8 | 29.2 | 8.5 | 8.0 |
| E2E (nov.) | 32.3 | 31.9 | 11.4 | 10.8 |
| E2E (rec.) | 32.8 | 32.6 | 10.5 | 10.0 |
| Ours (cov.) | **39.5** | **39.0** | 14.8 | 14.3 |
| Ours (sem.) | 37.7 | 37.4 | **16.0** | **15.4** |
| Ours (obs.) | 38.2 | 37.8 | 15.8 | 15.3 |
| Ours (view.) | 39.0 | 38.6 | 15.9 | 15.3 |

Table 4: **Planning performance** using the *Fast Marching* method on the BEV maps estimated from NeRFs.

| Policy | Conv. rate | Rot. Error (°) | Trans. Error (m) |
|---|---|---|---|
| Frontier | 7.2 | 0.383 | 0.00955 |
| E2E (cov.) | 15.4 | 0.319 | 0.00775 |
| E2E (cur.) | 12.5 | 0.325 | 0.00799 |
| E2E (nov.) | 19.4 | 0.315 | 0.00774 |
| E2E (rec.) | 19.3 | 0.292 | 0.00734 |
| Ours (cov.) | 20.2 | **0.283** | **0.00734** |
| Ours (sem.) | **23.0** | 0.319 | 0.00784 |
| Ours (obs.) | 22.5 | 0.305 | 0.00765 |
| Ours (view.) | 21.1 | 0.316 | 0.00769 |

Table 5: **Pose Refinement** – optimizing camera viewpoints given a rendered target viewpoint.

Figure 8: **Quality of semantic rendering** on pairs of images of different scenes, compared with GT from Sim. Policy: *Ours (obs)*.

| Task | Metrics | Sim. | MR-CNN |
|---|---|---|---|
| **Rendering** | Acc. | 91.8 | 65.4 |
| | mIoU | 83.2 | 61.1 |
| **Map comp.** | Acc. | 99.8 | 99.7 |
| | Prec. | 36.2 | 14.1 |
| | Rec. | 29.8 | 8.5 |
| **ObjGoal** | Succ. | 15.8 | 6.8 |
| | SPL | 15.3 | 6.5 |

Table 6: **NeRF semantic maps** – impact of the choice of ground-truth semantics vs. Mask R-CNN predictions using data collected by *Ours (obs.)*.

*tion* seem to be the best reward functions when training end-to-end policies if the final goal is to autonomously collect data to build a NeRF model.

**Comparing trained policies —** Rewarding modular policies with obstacles (*Ours (obs.)*) and viewpoints (*Ours (view.)*) coverage appears to lead to the best overall performance when we consider the different metrics. Explored area coverage (*Ours (cov.)*) leads to highest *PointNav* performance, corroborating its importance for geometric tasks, whereas other semantic reward functions lead to higher *ObjectNav* performance, again corroborating its importance for semantic understanding of the scene.

**Semantics from Mask R-CNN —** Table 6 shows the impact of using Mask R-CNN to compute the semantics training data of the NeRF model vs semantics from simulation. As expected, performance drops because Mask R-CNN provides a much noisier training signal, which could partly be explained by the visual domain gap between the real world and simulators. However, performance on the different downstream tasks is still reasonable, showing that one could autonomously collect data and generate semantics training signal without requiring additional annotation.

## 5.4 QUALITATIVE RESULTS

**BEV maps —** Figure 10 gives examples of the BEV maps generated from the continuous representation: structural details and dense semantic information are nicely recovered (Left). Planned trajectories are close to the shortest paths, for both PointGoal tasks (Middle) and ObjectGoal (Right).

**Semantic rendering —** Figure 9 compares the segmentation maps and RGB frames rendered with the continuous representation (trained with semantic masks from simulation) to the GT maps from the simulator. Again, the structure of the objects and even fine details are well recovered, and only very local noise is visible in certain areas. The semantic reconstruction is satisfying.

## 6 CONCLUSION

This work introduces a task involving navigating in a 3D environment to collect NeRF training data. We show that RL-trained modular policies outperform classic Frontier Based Exploration as well as other end-to-end RL baselines on this task, and compare different training reward functions. We also suggest evaluating NeRF from a scene-understanding point of view and with robotics-oriented tasks: BEV map generation, planning, rendering, and camera pose refinement. Finally, we show that it is possible with the considered method to reconstruct house-scale scenes. Interesting future work could target fine-tuning navigation models automatically on a scene.

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

# Appendix

## A    STUDY OF CORRELATIONS BETWEEN REWARDS AND DOWNSTREAM TASKS PERFORMANCE

We analyze the correlation between cumulated reward, used as a metric, and NeRF evaluation metrics. For each reward definition, we compute cumulated reward (*cov.*, *sem.*, *obs.*, *view.*) after 1500 timesteps. 5 runs for each policy on each scene provide 20 data points for each reward function on each scene. In Table 7 we report the Pearson correlation coefficient between the reward function and NeRF evaluation metric for the 5 Gibson val scenes if the associated p-value is lower than 5%, otherwise "−". As can be seen, obstacle coverage is the most correlated to NeRF evaluation, followed by viewpoints coverage.

## B    AUTONOMOUS ADAPTATION TO A NEW SCENE

We provide additional details regarding episode sampling and training hyperparameters when finetuning the pre-trained PointGoal policy on AutoNeRF and Gibson original meshes. Sampling of training and validation episodes, as well as training and validation, are performed in the Habitat simulator. For each environment, we sample $50k$ episodes from the mesh to finetune the policy for $10M$ training frames using PPO. The chosen learning rate was $2.5e-6$ and $2.5e-5$ when finetuning on AutoNeRF meshes and original Gibson meshes respectively. For evaluation, we sample $1k$ episodes per scene on the original Gibson meshes and report mean Success and SPL.

## C    NAVIGATING WITH SENSOR NOISE

All experiments in this work were conducted following task specifications in Chaplot et al. (2020c), among which are perfect odometry information and actuation. We thus conduct an additional experiment to evaluate the impact of sensor and actuation noise on AutoNeRF. We add noise using realistic models from Chaplot et al. (2020b) and correct odometry information using the pose estimation module trained in Chaplot et al. (2020b). This allows our modular policy (*Ours (obs.)*) to explore properly environments and collect NeRF training data. We then refine camera poses with bundle adjustment before using them to train NeRF models. Tables 8, 9, 10, 11 show that performance obviously decreases when adding noise, but we still reach a satisyfing performance in all metrics. It is also important to note that, even after using the pose estimation module and postprocessing bundle adjustment, camera poses spanning such large scenes are still noisy, and training NeRF models on noisy poses is considered a challenging problem in the literature. Better performance might thus come from new techniques to make NeRF models more robust to camera pose noise, which is orthogonal to the contribution in this work.

## D    GENERALIZATION TO ANOTHER NeRF VARIANT

To show that AutoNeRF can be generalized to another NeRF variant, we train an *Instant-NGP* (Müller et al., 2022) model on data collected by the modular policy (*Ours (obs.)*) and compare its rendering performance with the one of *Semantic Nerfacto* in Table 12. As can be seen, rendering performance is close.

## E    NAVIGATING INSIDE A NeRF-GENERATED MESH

To further evaluate the quality of the geometry learnt by an autonomously generated NeRF and assess to what extent it can be used inside a simulator, we perform PointGoal navigation on an original mesh and a NeRF-generated one for 4 Gibson val scenes. The considered agent is based on the modular policy introduced in the main paper, restricted to the planning part: the output of the Global Policy is replaced with the input PointGoal vector. Planning is performed using the Fast Marching Method, relying on the map channels storing information about obstacles and explored area. Table 13 shows that there is a performance drop between the navigating on the original Gibson meshes

| | PSNR (RGB rendering) | | | | | Per-class acc (Sem rendering) | | | | |
|---|---|---|---|---|---|---|---|---|---|---|
| Reward | 0 | 1 | 2 | 3 | 4 | 0 | 1 | 2 | 3 | 4 |
| cov. | 0.73 | 0.77 | 0.95 | 0.48 | – | – | 0.59 | **0.97** | 0.60 | – |
| sem. | 0.52 | – | 0.84 | 0.62 | – | – | 0.66 | 0.82 | 0.61 | – |
| obs. | **0.88** | **0.83** | **0.96** | **0.70** | – | **0.48** | **0.79** | 0.95 | **0.76** | 0.48 |
| view. | 0.83 | 0.55 | 0.93 | 0.67 | **0.55** | – | 0.48 | 0.87 | 0.70 | **0.55** |

| | Occ. recall (Map comp.) | | | | | Sem. recall (Map comp.) | | | | |
|---|---|---|---|---|---|---|---|---|---|---|
| Reward | 0 | 1 | 2 | 3 | 4 | 0 | 1 | 2 | 3 | 4 |
| cov. | 0.50 | 0.75 | **0.94** | 0.47 | – | – | 0.47 | **0.93** | – | – |
| sem. | 0.53 | 0.60 | 0.80 | 0.48 | 0.46 | – | **0.73** | 0.72 | – | – |
| obs. | **0.73** | **0.87** | **0.94** | **0.56** | 0.64 | – | 0.72 | 0.91 | – | – |
| view. | 0.66 | – | 0.88 | 0.53 | **0.69** | – | 0.60 | 0.80 | – | – |

| | PointGoal Succ (Planning) | | | | | ObjectGoal Succ (Planning) | | | | |
|---|---|---|---|---|---|---|---|---|---|---|
| Reward | 0 | 1 | 2 | 3 | 4 | 0 | 1 | 2 | 3 | 4 |
| cov. | – | – | 0.63 | – | – | – | – | 0.62 | – | – |
| sem. | – | – | – | – | – | – | – | 0.44 | – | – |
| obs. | – | – | **0.66** | – | – | – | – | **0.66** | – | – |
| view. | – | – | 0.60 | – | **0.53** | – | – | 0.60 | – | **0.53** |

| | Rot. conv. rate (Pose Ref.) | | | | | Trans. conv. rate (Pose Ref.) | | | | |
|---|---|---|---|---|---|---|---|---|---|---|
| Reward | 0 | 1 | 2 | 3 | 4 | 0 | 1 | 2 | 3 | 4 |
| cov. | 0.49 | – | 0.69 | – | – | 0.54 | – | 0.72 | – | – |
| sem. | 0.67 | – | 0.70 | – | – | 0.52 | – | 0.73 | **0.63** | 0.49 |
| obs. | 0.69 | – | 0.72 | – | 0.59 | 0.66 | – | 0.75 | 0.52 | 0.65 |
| view. | **0.74** | – | **0.75** | – | **0.66** | **0.70** | – | **0.78** | 0.60 | **0.81** |

**Table 7: Correlations between reward metrics and selected NeRF evaluation metrics**. Pearson correlation coefficients are reported if the associated p-value is lower than $5\%$, otherwise $-$. Columns denoted 0, 1, 2, 3, 4 correspond to the 5 Gibson val scenes. Obstacle coverage is the most correlated to NeRF evaluation metrics, followed by viewpoints coverage.

| | | RGB | | | Semantics | |
|---|---|---|---|---|---|---|
| Policy | Noise | PSNR | SSIM | LPIPS | Acc | mIoU |
| **Ours (obs.)** | – | 25.56 | 0.846 | 0.203 | 91.8 | 83.2 |
| **Ours (obs.)** | ✓ | 20.87 | 0.761 | 0.264 | 85.4 | 73.6 |

**Table 8:** Impact of noise on rendering

| | | Occupancy | | | Semantics | | |
|---|---|---|---|---|---|---|---|
| Policy | Noise | Acc. | Prec. | Rec. | Acc. | Prec. | Rec. |
| **Ours (obs.)** | – | 86.4 | 89.4 | 76.5 | 99.8 | 36.2 | 29.8 |
| **Ours (obs.)** | ✓ | 86.8 | 89.8 | 69.6 | 99.7 | 29.9 | 24.1 |

**Table 9:** Impact of noise on map estimation

| | | PointGoal | | ObjectGoal | |
|---|---|---|---|---|---|
| Policy | Noise | Succ. | SPL | Succ. | SPL |
| **Ours (obs.)** | – | 38.2 | 37.8 | 15.8 | 15.3 |
| **Ours (obs.)** | ✓ | 34.5 | 33.8 | 12.9 | 12.4 |

**Table 10:** Impact of noise on planning

| Policy | Noise | Conv. rate | Rot. Error ($^\circ$) | Trans. Error (m) |
|---|---|---|---|---|
| **Ours (obs.)** | – | 22.5 | 0.305 | 0.00765 |
| **Ours (obs.)** | ✓ | 7.9 | 0.405 | 0.01125 |

**Table 11:** Impact of noise on pose refinement

and the reconstructed ones, but Success and SPL are close. The performance on the original mesh is a "soft upper bound", as data was collected from navigating in this original mesh, before training a NeRF and finally generating a new mesh representation. These results show that our NeRF-generated mesh features a satisfying geometry, allowing to navigate properly when loaded within the Habitat simulator. Some further mesh post-processing could be needed, along with additional work on improving lighting within the simulator.

# F    DETAILS ON DOWNSTREAM TASKS

**Metric Map Estimation —** Cells in the occupancy and semantic top-down maps generated from NeRF models are of size $1cm \times 1cm$. In order to create the occupancy map, we first compute a 3D voxel grid by regularly querying the NeRF density head between scene bounds along $x$ and $z$ axes, and between 0 and the agent's height along the vertical $y$ axis. We then transform the 3D grid into a 2D top-down map by applying a sum operation along the vertical axis. We found that, when using a *Semantic Nerfacto* model, generating a point cloud where each point is associated with a semantics class from the train camera poses works best when generating the semantics top-down map. The point cloud is converted into a 3D voxel grid, where each cell is associated with a channel for each semantics class. A per-class sum operation can finally transform the 3D grid into a 2D map with one channel per class. The same cell resolution is used when generating ground-truth maps from the Habitat simulator.

| NeRF model | PSNR | SSIM | LPIPS |
|---|---|---|---|
| **Semantic Nerfacto** | 25.56 | 0.846 | 0.203 |
| **Instant-NGP** | 24.92 | 0.806 | 0.279 |

**Table 12:** Generalizing to another NeRF variant

| Mesh | Success | SPL |
|---|---|---|
| **Original Gibson mesh** | 88.1 | 73.3 |
| **Rollout $\rightarrow$ NeRF training $\rightarrow$ Mesh generation** | 78.4 | 67.7 |

**Table 13: Navigating inside a NeRF-generated mesh**: Average PointGoal performance of a policy planning a path to the goal with the Fast Marching Method and taking discrete actions on 4 Gibson val scenes in the Habitat simulator, from either the original mesh (**Gibson**) or the mesh extracted from the NeRF model (**Rollout + NeRF train. + Mesh gen.**) trained from autonomously collected data by *Ours (obs.)*. Our reconstruction does not require depth data.

**Planning —** Resolution of the top-down maps used to plan a path are the same as for the *Metric Map Estimation* task. Once generated, the path is evaluated on the ground-truth top-down map from the Habitat simulator. In order to account for the size of a potential robot of radius $18cm$, obstacles on the ground-truth map are dilated. We also apply a dilation with a $20cm$ radius to obstacles on our top-down map before planning the path.

For a given episode $i$, *Success* $S_i$ is 1 if the last cell in the planned path is closer than 1m to the goal and if less than 10 planned cells are obstacles, otherwise it is 0. We chose to allow up to 10 obstacle cells on the planned path to keep the task from being overly complex and thus uninformative.

For both *PointGoal planning* and *ObjectGoal planning*, we report mean *Success* and *SPL* over a total of $N$ planning episodes. Mean *Success* is $\frac{1}{N}\sum_{i=1}^{N} S_i$. Mean *SPL* takes into account both success and path efficiency to reach the goal and is equal to $\frac{1}{N}\sum_{i=1}^{N} S_i \frac{\ell_i}{\max(p_i, \ell_i)}$, where $\ell_i$ if the shortest path distance from the start point to the goal and $p_i$ is the length of the planned path.

**Pose Refinement —** At each pose refinement optimization step, we would ideally want to render the full image associated with the estimated camera pose to compare with the RGB frame from the camera. As already noticed in previous work (Yen-Chen et al., 2021), doing this is expensive, and we thus instead randomly sample pixels to be rendered within the image at each optimization step.

**Mesh generation —** In order to create a mesh from a trained NeRF model, we first build a 3D voxel grid by querying the implicit representation regularly on a grid between the scene bounds. Each voxel will be associated with a density, and either a color or a semantics class depending on the nature of the mesh to generate. The voxel grid is converted into a mesh by applying the Marching Cubes algorithm (Lorensen & Cline, 1987).

# G  QUALITATIVE EXAMPLES

We provide additional qualitative results for the rendering, map estimation, planning, pose refinement and mesh generation tasks.

**Rendering —** Figure 9 shows additional rendering examples from a *Semantic Nerfacto* model trained on data collected by the modular policy, *Ours(obs.)*, with semantics ground-truth from the Habitat simulator. Both RGB and semantics rendering are accurate for camera poses that were not seen during NeRF training.

**Metric Map Estimation and Planning —** Figure 10 shows additional results regarding semantic top-down map generation and path planning (both *PointGoal* and *ObjectGoal*) from a *Semantic Nerfacto* model trained on data collected by the modular policy, *Ours(obs.)*, with semantics ground-truth from the Habitat simulator. As done in Figure 7 in the main paper, we present qualitative results from a *Semantic Nerfacto* model as it is the one we use in quantitative results (Tables 2, 3, 4, 5 in the main paper). Higher-quality geometry could be obtained from a vanilla *Semantic NeRF* model, but with the cost of significantly longer training time. Semantic objects are properly localized, geometry is correct, except for some room corners that are more challenging to properly reconstruct with the fast-trained *Semantic Nerfacto*. Paths can be planned to both end points specified as positions on the grid and to the closest object of a given category.

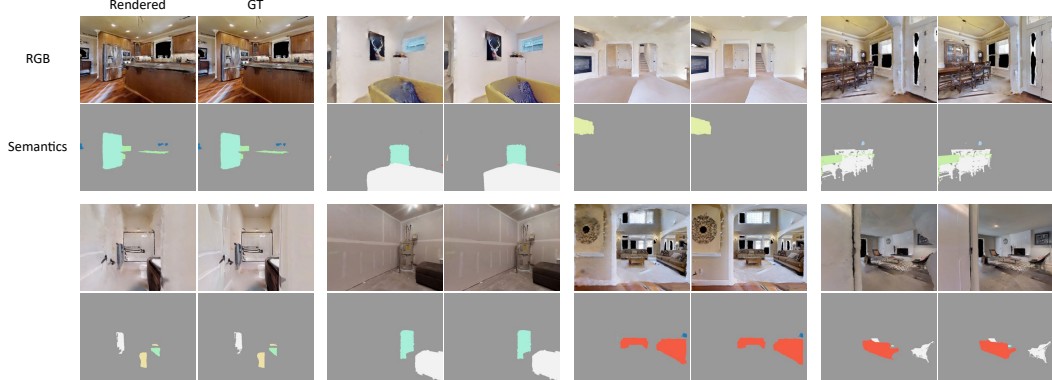

**Figure 9: Quality of semantic rendering** on pairs of images of different scenes, compared with GT from Sim. NeRF training data is collected by *Ours (obs)*.

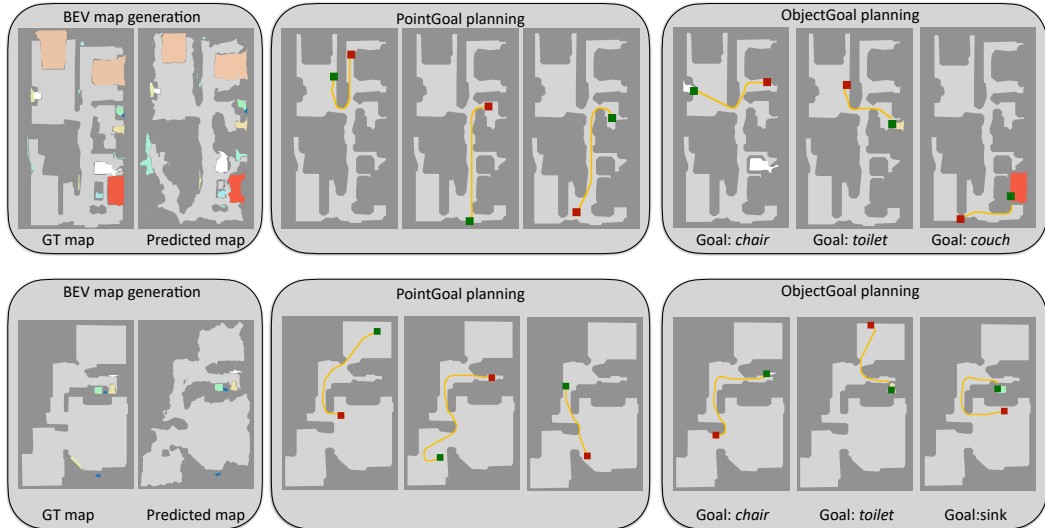

**Figure 10: BEV map tasks**: Generation of semantic BEV maps (Left), *PointGoal* (Middle) and *ObjectGoal* planning (Right). NeRF training data is collected by *Ours (obs)*.

**Pose refinement —** Camera pose optimization results are best viewed as videos. Examples are available in the Supplementary Material, showcasing the evolution of *Semantic Nerfacto* rendered frame compared with the ground-truth camera view during the optimization process. The NeRF was trained on data collected by the modular policy, *Ours(obs.)*.

**Mesh Generation —** Figure 11 shows the RGB and semantics meshes extracted from a *vanilla Semantic NeRF* model trained on data collected by the modular policy, *Ours(obs.)*, with semantics ground-truth from the Habitat simulator. The RGB mesh is close to the ground-truth original Gibson mesh, while being built without any depth input. Objects of interest are properly segmented on the semantics mesh.

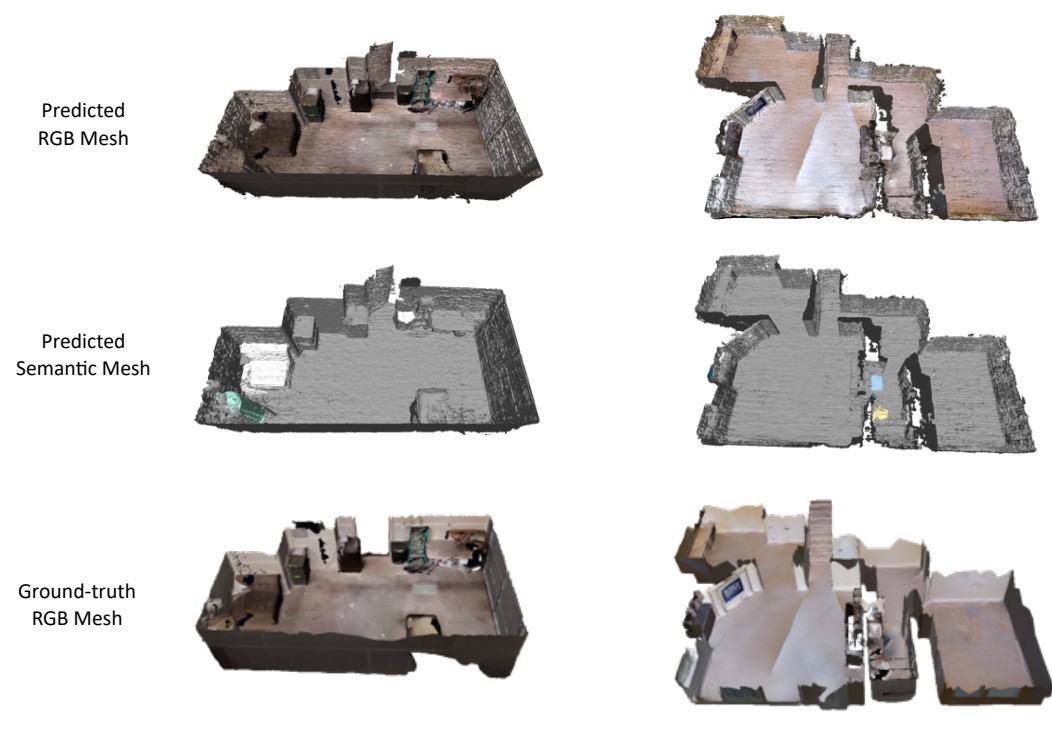

Predicted
RGB Mesh

Predicted
Semantic Mesh

Ground-truth
RGB Mesh

Scene name    **Collierville**                    **Wiconisco**

**Figure 11: Mesh reconstruction**: reconstruction of 2 Gibson val scenes extracted from a NeRF model trained on data gathered by our *Ours (obs)* modular policy. Both geometry, semantics, and appearance are satisfying.

