# OpenReview forum: "AutoNeRF: Training Implicit Scene Representations with Autonomous Agents"
_ICLR.cc/2024/Conference — Submitted to ICLR 2024_

### Official Review · Reviewer_ZNfM · 2023-10-29

**Soundness:** 2 fair
**Presentation:** 3 good
**Contribution:** 2 fair
**Rating:** 5
**Confidence:** 4

**Summary:**

The paper targets an important problem of autonomous NeRF construction through active exploration. The solution is straightforward: A conventional modular learning of online exploration is first deployed, followed by an offline semantic NeRF construction. It should be noted that the two types of scene representations are adopted for these two problems. Multiple downstream tasks are conducted given the batch-learned NeRF.

**Strengths:**

+ The paper is well-organized and clearly presented. The motivation is clear and the problem setting is practical for the NeRF community.
+ Multiple downstream tasks given the trained NeRF are tested.
+ The utilization of a NeRF model as a simulation environment for finetuning navigation policy is interesting.

**Weaknesses:**

1. The major concern lies in the two-phase fashion:
- The input of the system is unclear. For exploration, RGB frames along with the depth maps are required (Sec. 4 Task Specification), while for NeRF training, the depth information is not leveraged. It is highlighted in Sec. 4.2 that 'no depth information is required for reconstruction' is an important property. From the reviewer's perspective, this is merely the characteristic of NeRF itself. For the proposed AutoNeRF, the depth information is essential for the proposed exploration module to update the 2D top-down map.
- How the automatic process helps the downstream tasks is unclear. It is known that a trained NeRF can be applied for these four tasks (as also mentioned in the Related Work section). The paper directly adopts the methods without further modification, and the NeRF training does provide any feedback to the exploration policy training. Moreover, the exploration phase already builds a 2D map online that can be used for Task 2&3. The benefit of involving an extra NeRF training phase for these two tasks should be further elaborated.

2. Insufficient experiments.
- Though the paper conducts thorough experiments on multiple tasks, merely 5 Gibson scenes are evaluated. It should be noted that the methods of [Chaplot et al., 2020a] and [Ramakrishnan et al. (2021)] are evaluated on Matterport3D besides the Gibson datasets. As the paper targets an AutoNeRF problem, the method is expected not to be limited by the data-driven exploration policy. The generalization ability of the proposed modular learning strategy should be discussed compared to the relevant one-phase works of A-C.
- The technical contributions seem to be the different reward signals in Sec. 4.1. Detailed ablation studies should be conducted to validate the efficacy and the benefits arising from these terms.

A. Yunlong Ran et al., Neurar: Neural uncertainty for autonomous 3d reconstruction with implicit neural representations. RA-L 2023
B. Jing Zeng et al., Efficient view path planning for autonomous implicit reconstruction. ICRA 2023
C. Zike Yan et al., Active neural mapping. ICCV 2023

**Questions:**

Though the automatic reconstruction of the implicit neural field is of great importance, the current version seems inconsistent and disintegrated. Multiple aspects are integrated into the system directly without clear connections. The author is expected to stress the following questions in a more unified and valid manner:
1. What is the major focus of the paper? A. If the paper targets an automatic reconstruction of the neural radiance field, there are only minor contributions to the reward design of the exploration policy. The author should further evaluate how these rewards lead to a better radiance field (but not a better exploration strategy). The limitations of this modular learning strategy should also be discussed (especially the generalization ability). B. If the paper targets a better exploration policy with additional NeRF training, how NeRF affects the exploration policy (but not the pointGoal policy provided in Tab. 1) should be elaborated. Nevertheless, it seems that the construction of NeRF does not affect the exploration policy in this modular fashion. C. If the paper aims to stress the advantages of a well-trained NeRF given sufficient observations on the downstream tasks, the experiments should clearly explain how the compared exploration strategies (Frontier, E2E) fail that leads to performance degradation, and how the designed reward functions help to solve the problem.
2. What is the role of semantic reasoning? Seemingly the semantic cues are not the best reward for exploration, and semantics are not a prerequisite for the 'AutoNeRF' setting. The subtasks of semantic rendering and object-goal navigation can be removed without hurting the central contribution. The role of semantics should be better demonstrated.
3. Does the training of NeRF lead to a better BEV map? The exploration phase already obtains a 2D metric-semantic map on the fly that can be applied for exploration and point/object-goal navigation. The map should be added to Fig. 10 for a better understanding of how the performance can be affected with a follow-up NeRF model.

---

> ### Author Response · Authors · 2023-11-21
> **Answer to Reviewer ZNfM (Part 1)**
>
> > The input of the system is unclear. For exploration, RGB frames along with the depth maps are required (Sec. 4 Task Specification), while for NeRF training, the depth information is not leveraged. It is highlighted in Sec. 4.2 that 'no depth information is required for reconstruction' is an important property. From the reviewer's perspective, this is merely the characteristic of NeRF itself. For the proposed AutoNeRF, the depth information is essential for the proposed exploration module to update the 2D top-down map.
>
> The exploration policy indeed requires depth input. The lack of required depth information at  reconstruction time has different implications:
> 1. This is a quality of NeRF models, motivating the interest in such 3D scene representation and thus to focus on how to autonomously collect NeRF training data.
> 2. Previous work (*Gervet et al., Navigating to Objects in the Real World, Science Robotics, 2023*) has shown that RL-trained modular policies can be deployed on real robots with low-resolution noisy depth sensors. However, in such a scenario, while the navigation policy will be able to properly explore the environment, a low-resolution noisy depth input might lead to poor 3D reconstruction with methods relying on depth, showing the interest of our framework. We could have a policy to navigate properly from low-quality depth and, in a second stage, obtain satisfying reconstruction from NeRF that does not depend on depth sensing quality.
> 3. Finally, the fact that such NeRF models do not require depth input is highly dependent on the quality of the underlying collected data. Without a proper diversity of camera frames and viewpoints, NeRF training is likely to fail. By following standard practices of training NeRF models without depth input, we increase the importance of the active data collection part in the final NeRF model performance, allowing to better evaluate considered baselines.
>
> > How the automatic process helps the downstream tasks is unclear. It is known that a trained NeRF can be applied for these four tasks (as also mentioned in the Related Work section). The paper directly adopts the methods without further modification, and the NeRF training does provide any feedback to the exploration policy training. Moreover, the exploration phase already builds a 2D map online that can be used for Task 2&3. The benefit of involving an extra NeRF training phase for these two tasks should be further elaborated.
>
> Downstream tasks are used in this work as an evaluation protocol of the underlying quality of a given NeRF model. While many recent works focus on evaluating NeRFs through novel view synthesis, we consider a suite of tasks that allow a more thorough evaluation. Among these tasks is occupancy map estimation from the NeRF representation which allows to verify the geometric understanding of the 3D scene by the NeRF model.
>
> > Though the paper conducts thorough experiments on multiple tasks, merely 5 Gibson scenes are evaluated. It should be noted that the methods of [Chaplot et al., 2020a] and [Ramakrishnan et al. (2021)] are evaluated on Matterport3D besides the Gibson datasets. As the paper targets an AutoNeRF problem, the method is expected not to be limited by the data-driven exploration policy. The generalization ability of the proposed modular learning strategy should be discussed compared to the relevant one-phase works of A-C.
>
> All baselines in this work are evaluated on 5 validation scenes that are different from considered scenes at training time.
>
> The difference with previous work can be easily explained when considering the time involved in evaluating one agent episode on a new scene: in previous work ([Chaplot et al., 2020a] and [Ramakrishnan et al. (2021)]), such an evaluation simply involves neural network forward passes and simulation renderings. All of this can be done very rapidly, leading to an evaluation time of a few seconds for a given episode (with the ability to parallelize to have even faster evaluation). In our case, evaluating one episode involves navigation and simulation like previous work, but also training a NeRF model for this specific episode in order to evaluate its quality. Semantic Nerfacto takes up to 30 minutes to train in large house-scale scenes, which means that one agent rollout takes 30 minutes to evaluate against seconds for previous work. We have already put a lot of computation into our evaluation protocol in order to assess the generalization abilities of all the proposed agents.

---

> > ### Author Response · Authors · 2023-11-21
> > **Answer to Reviewer ZNfM (Part 2)**
> >
> > > The technical contributions seem to be the different reward signals in Sec. 4.1. Detailed ablation studies should be conducted to validate the efficacy and the benefits arising from these terms.
> >
> > Our paper compares the impact of training the modular policy with the different reward functions, showcasing the gains brought by different ones on the considered downstream tasks. Please refer to our global answer to all reviewers for details about conclusions that can be drawn from our experimental study. Tables 2-5 in the main paper compare the performance of modular policies trained with the different reward functions.
> >
> > > What is the major focus of the paper?
> >
> > Please refer to our global answer to all reviewers.
> >
> > > What is the role of semantic reasoning? Seemingly the semantic cues are not the best reward for exploration, and semantics are not a prerequisite for the 'AutoNeRF' setting. The subtasks of semantic rendering and object-goal navigation can be removed without hurting the central contribution. The role of semantics should be better demonstrated.
> >
> > Semantics is closely tied to geometry. Properly covering the whole scene with diverse viewpoints and camera positions leads to a better geometric representation of a 3D scene. Being able to map semantic information requires a good knowledge of geometry, which makes semantic rendering a relevant task to evaluate NeRF quality. Moreover, semantic mapping is important from an applicative point of view if we consider the final NeRF model as a persistent representation of a scene of interest.
> >
> > > Does the training of NeRF lead to a better BEV map? The exploration phase already obtains a 2D metric-semantic map on the fly that can be applied for exploration and point/object-goal navigation. The map should be added to Fig. 10 for a better understanding of how the performance can be affected with a follow-up NeRF model.
> >
> > As already mentioned in a previous part of our answer, downstream tasks are used in this work as an evaluation protocol of the underlying quality of a given NeRF model. The main point of evaluating the BEV map derived from the NeRF model is to assess the quality of its geometric understanding.

---

### Official Review · Reviewer_6sGZ · 2023-11-01

**Soundness:** 3 good
**Presentation:** 3 good
**Contribution:** 3 good
**Rating:** 6
**Confidence:** 3

**Summary:**

The paper introduces AutoNeRF, a method enabling autonomous agents to collect data for training NeRF without human intervention. It proposes various exploration strategies for efficient environment exploration. These strategies are evaluated based on downstream tasks like viewpoint rendering, map reconstruction, planning, and pose refinement. Results demonstrate that NeRFs can be trained from a single episode in an unknown environment and applied to diverse robotic tasks.

**Strengths:**

1. The paper explores multiple exploration policies for collecting training samples for a scene NERF, which provides a comprehensive analysis of the community.
2. The policies are further evaluated using different downstream robotic tasks, which is beneficial to related researchers.
3. The experimental results are comprehensive and solid.
4. The paper is well-written and easy to follow.

**Weaknesses:**

1. The novelty of the paper is limited. The idea of using Nerf for scene construction in SLAM is not new. The authors also adopted an off-the-shelf NERF module. The main contribution of this paper lies in validating the effect of different exploration policies during image collection on downstream robotics tasks.

**Questions:**

1. How many images are used to train the NERF model? In task specification it says "The agent can navigate for a limited number of 1500 discrete steps", does it mean to capture 1500 images for NERF training?

---

> ### Author Response · Authors · 2023-11-21
> **Answer to Reviewer 6sGZ**
>
> > The novelty of the paper is limited. The idea of using Nerf for scene construction in SLAM is not new. The authors also adopted an off-the-shelf NERF module. The main contribution of this paper lies in validating the effect of different exploration policies during image collection on downstream robotics tasks.
>
> Please refer to our global answer to all reviewers for a detailed description of the contributions of this paper.
>
> > How many images are used to train the NERF model? In task specification it says "The agent can navigate for a limited number of 1500 discrete steps", does it mean to capture 1500 images for NERF training?
>
> The agent can indeed navigate for 1500 steps, leading to a training set that is made of up to 1500 training samples. However, we perform a filtering to remove images associated with close camera poses. In practice, when deploying the modular policy trained with the *obstacle coverage reward* on previously unseen scenes, the mean number of collected images is 754, with a minimum of 264 in the smallest scene and a maximum of 1257 in the largest scene.

---

### Official Review · Reviewer_7hgK · 2023-11-01

**Soundness:** 3 good
**Presentation:** 2 fair
**Contribution:** 3 good
**Rating:** 5
**Confidence:** 3

**Summary:**

The paper discusses the key considerations when employing NeRF for map construction and semantic segmentation in the context of autonomous embodied agents exploring previously unseen environments. Utilizing modular exploration policies that take into account four types of intrinsic rewards (exploration area, obstacle coverage, semantic object coverage, and viewpoint coverage), the agent initially learns how to explore through self-supervised learning. Subsequently, the agent trains a Semantic Nerfacto by collecting valuable data during its exploration. The quality of the Nerfacto is assessed across four distinct tasks: rendering, metric map estimation, planning, and pose refinement. The authors propose reward types that enhance NeRF training and present a diverse set of evaluation tasks to expand the application of NeRF in the field of environment exploration.

**Strengths:**

The authors correctly emphasize the need for more realistic tasks when evaluating NeRF in navigation scenarios. Furthermore, they have introduced effective rewards that yield improved NeRF results in the various suggested tasks.

**Weaknesses:**

As the authors attempt to cover a wide range of tasks, the presentation of results lacks organization, making it challenging for readers to interpret and examine the outcomes. Further weaknesses are outlined in the "Questions" section.

**Questions:**

1. Could you please share the results of the vanilla Semantic NeRF model mentioned in Figures 9 and 10? While it is claimed to deliver superior results compared to Semantic Nerfacto, there is a lack of supporting evidence.
2. It appears that the four types of rewards exhibit different strengths. Are there any criteria or trends that can assist researchers in selecting the most appropriate reward type for specific tasks? For instance, is there a reason why the policy with the obstacle coverage reward performs better than others in the rendering task?

3. Could you provide a more detailed explanation of the experiments featured in Table 1 and elaborate on their implications?

---

> ### Author Response · Authors · 2023-11-21
> **Answer to Reviewer 7hgK**
>
> > As the authors attempt to cover a wide range of tasks, the presentation of results lacks organization, making it challenging for readers to interpret and examine the outcomes.
>
> The different considered downstream tasks allow to more thoroughly evaluate the performance of the policy in its ability to collect training data that will lead to high-quality NeRF representations, which is one of the contributions of this work. This is in stark contrast to a large body of recent work which solely focuses on novel view synthesis, while we attempt to go one step further and focus on concrete applications in robotics. Please refer to our global answer to all reviewers for more details about the contributions of this work and the conclusions drawn from experiments.
>
> > Could you please share the results of the vanilla Semantic NeRF model mentioned in Figures 9 and 10? While it is claimed to deliver superior results compared to Semantic Nerfacto, there is a lack of supporting evidence.
>
> The *vanilla Semantic NeRF* model leads to a higher-quality 3D mesh after applying our mesh extraction protocol. The difference was assessed qualitatively by witnessing holes and irregularities in the *Semantic Nerfacto* final mesh. The latter still provides satisfying results as can be seen quantitatively and with qualitative BEV maps, and is much faster to train, but our aim here was to show the best practical results in terms of final 3D scans one could get when applying AutoNeRF.
>
> > It appears that the four types of rewards exhibit different strengths. Are there any criteria or trends that can assist researchers in selecting the most appropriate reward type for specific tasks? For instance, is there a reason why the policy with the obstacle coverage reward performs better than others in the rendering task?
>
> Different reward functions can lead to diverse policy behaviors, which is why we report the performance of the different modular policies on all the downstream tasks. The chosen reward might depend on the specific considered task, but in this work we were rather interested in finding the reward function that leads to the best overall performance on the different downstream tasks. From collected results, obstacle coverage and viewpoints coverage lead to the best overall performance.
>
> In this work, the rendering ability of a NeRF model is evaluated by sampling camera poses in the scene and comparing NeRF rendering with rendering from the simulator. The difficulty of the task is thus linked to rendering different objects that might not have been observed thoroughly, while redundant parts of the scene such as floors, walls, ceilings have likely been observed from different viewpoints by any policy with ease. This explains the high performance of the obstacle coverage reward, as it guides the policy to focus on covering specific objects or obstacles in the scene.
>
> > Could you provide a more detailed explanation of the experiments featured in Table 1 and elaborate on their implications?
>
> Table 1 presents an application of AutoNeRF to a practical scenario where a user would have bought a brand new robot that navigates using a neural policy (different from AutoNeRF) that was pre-trained in simulation. They now want to use the robot in their house, but even if the robot policy was pre-trained on a large and diverse set of scenes, it is quite likely it will struggle with some specificities of their house. A solution to this would be scene-specific fine-tuning: fine-tuning the robot policy on their house. However, such fine-tuning must be performed safely so a good thing would be to do it in simulation. A question is thus: can we use AutoNeRF to autonomously explore a new scene, collect data and train a NeRF? We can then extract a mesh representation from the trained NeRF, load this mesh into a simulator and fine-tune the robot policy of interest. Table 1 shows results of similar experiments performed in simulation: we show that such a procedure allows to improve the performance of a PointGoal agent, pre-trained on diverse scenes, by fine-tuning it on previously unseen scenes scanned by AutoNeRF.

---

### Official Review · Reviewer_o9wx · 2023-11-08

**Soundness:** 2 fair
**Presentation:** 2 fair
**Contribution:** 1 poor
**Rating:** 3
**Confidence:** 2

**Summary:**

This paper introduces a method, AutoNeRF, that enables an embodied agent to autonomously collect the necessary data to train Neural Radiance Fields (NeRF), thereby removing the need for manual and tedious data collection. The authors compared various exploration strategies including frontier-based, end-to-end and modularized approaches which consist of trained high-level planning and traditional path followers. The evaluation is based on four downstream tasks - rendering, map reconstruction, planning, and pose refinement. The results reveal that NeRFs can be efficiently trained using actively acquired data from an unseen environment, and that these can be used for different robotic tasks. Furthermore, they also show that modularly trained exploration models are superior to classical and end-to-end strategies.

**Strengths:**

1. The proposed autonomous approach for embodied agents collecting NeRF visual training data greatly reduces human intervention.
2. The authors conduct a comprehensive evaluation assessing the quality of the reconstructed scene produced by four different autonomous data collecting approaches. The assessment is carried out by evaluating each approach based on four downstream tasks which indicates the actual performance on follow-up applications.

**Weaknesses:**

1. The scientific contribution of this paper is unclear. Although the authors conduct a comprehensive evaluation of the designed autonomous approaches, it does not point out their core contributions to this field. The proposed autonomous exploration strategy is straight-forward, and could not be considered as main contribution of this paper. Additionally, the absence of a comparison between their methods and existing techniques in autonomous visual exploration—which could have been utilized for collecting NeRF training data—is missing. The experimental evidence alone does not constitute significant scientific contribution.
2. The claims are backed by empirical results, but the lack of theoretical analysis or concrete mathematical justifications for the algorithm.

**Questions:**

Did you evaluate the end-to-end based and modularized based method on completed unseen (or largely different) scenario? I am curious about the actual performance in such scenarios. While the rule-based method, i.e., the frontier-based exploration, did not outperform learning-based methods (such as end-to-end based and modularized methods) in your experiments, it demonstrated fairly consistent results across the majority of scenes, regardless of whether they were familiar or unfamiliar scenarios.

---

> ### Author Response · Authors · 2023-11-21
> **Answer to Reviewer o9wx**
>
> > The scientific contribution of this paper is unclear. Although the authors conduct a comprehensive evaluation of the designed autonomous approaches, it does not point out their core contributions to this field. The proposed autonomous exploration strategy is straight-forward, and could not be considered as main contribution of this paper. Additionally, the absence of a comparison between their methods and existing techniques in autonomous visual exploration—which could have been utilized for collecting NeRF training data—is missing. The experimental evidence alone does not constitute significant scientific contribution.
>
> Please refer to our global answer to all reviewers (“Contributions of this work” section) for a detailed description of the contributions of this paper. As mentioned, our contributions are on the experimental side, and we do compare the introduced modular policies with state-of-the art end-to-end visual exploration methods (*Ramakrishnan et al. An exploration of embodied visual exploration, IJCV 2021*). We believe that autonomous NeRF training data collection is an important topic and, to the best of our knowledge, this is the first work to target this direction, providing diverse experimental results.
>
> > The claims are backed by empirical results, but the lack of theoretical analysis or concrete mathematical justifications for the algorithm.
>
> We follow practices in the Embodied AI domain, where contributions are evaluated experimentally by assessing the ability of introduced methods to properly navigate in previously unseen environments at test time. Indeed, all 3D scenes used at test time were not seen during training by any policy. We thus respectfully believe that such a theoretical analysis is out of the scope of this work, while being an interesting future direction to keep in mind.
>
> > Did you evaluate the end-to-end based and modularized based method on completed unseen (or largely different) scenario? I am curious about the actual performance in such scenarios. While the rule-based method, i.e., the frontier-based exploration, did not outperform learning-based methods (such as end-to-end based and modularized methods) in your experiments, it demonstrated fairly consistent results across the majority of scenes, regardless of whether they were familiar or unfamiliar scenarios.
>
> Please refer to our global answer to all reviewers (“Generalization to new scenarios” section) for a description of our experimental setup which applies to end-to-end and modular methods.

---

> > ### Comment · Reviewer_o9wx · 2023-11-23
> >
> > First, I would like to thanks for the authors' responses.
> >
> > After browsing all comments from the authors, I am not convinced by the responses. The core contributions of a paper cannot be solely determined by a series of experimental results. More profound insights regarding these results should be carefully discussed, and what we could learn from those failure cases. Additionally, further experiments (more baselines and evaluate them on different datasets) should be conducted if they constitute the majority of the contributions.
> >
> > I believe a standard train/test split for dataset evaluation may not sufficiently assess the generalization capability of the proposed method in this study. This approach lacks the necessary evidence for a comprehensive understanding. I recommend exploring the method's performance across diverse datasets, such as training on dataset A and testing on dataset B or in a real-world environment. Without such analysis, we cannot ascertain the actual robustness and performance of the proposed method. When considering real-world deployment, the current evaluation leaves us uncertain about its actual performance.
> >
> > I will keep my rating as it is.

---

### Author Response · Authors · 2023-11-21
**Global answer to all reveiwers**

We thank the reviewers for their valuable feedback on our work, and address three main points below. Individual answers are provided separately.


1. **Contributions of this work**:  This paper is experimental in nature and we sometimes see that this line of work is confused with methodological or theoretical papers, but we believe that the gains of knowledge are interesting to the field. To the best of our knowledge, this is the first work studying the autonomous collection of NeRF training data with an egocentric agent in house-scale scenes. We show for the first time that a modular RL-trained policy can achieve such a task, and compare its performance with state-of-the-art end-to-end visual exploration methods. Another important contribution of this paper is the proposed reflection about how to properly evaluate the performance of a trained NeRF model in the context of robotics: we consider many robotics-related downstream tasks that give a more complete picture than only focusing on novel view synthesis. We also explore applications of this work such as house-scale scene reconstruction, and more specifically, how AutoNeRF can be used to autonomously scan a 3D scene to then perform a safe fine-tuning adaptation in simulation.


2. **Generalization to new scenarios**: In this work, we follow practices in the Embodied AI community, by training all considered policies on a set of training scenes, and evaluating them on a held-out set of unknown 3D environments. All reported performance scores (on all considered metrics) thus showcase the ability to generalize to new environments at test time.


3. **Conclusions drawn from experiments**: The main conclusions that can be drawn from the conducted studies are that NeRFs cannot only be constructed from small scenes but also on realistic house-scale ones and, more importantly, that this can be done in a fully automatic way. We show a use-case closing the loop, using AutoNeRF to autonomously generate a 3D representation of a new scene to then have a policy to improve itself in the considered environment in simulation. Another interesting conclusion is that training modular policies with object coverage and viewpoints coverage as reward functions lead to the overall best performance across tasks, and that the introduced modular policies outperform end-to-end and classical exploration baselines on all metrics.

---

### Meta-Review · Area_Chair_usqU · 2023-12-08

**Metareview:**

This paper proposes AutoNeRF, that uses and embodied agent to autonomously collect the data to train Nerfs. The experiments consider various exploration techniques which are then evaluated on multiple downstream tasks like mapping, view synthesis, planning, and pose refinement etc.

Most of the reviewers (3/4) are not supportive for acceptance and one of the reviewers is mildy positive. The biggest criticism is in terms of technical contribution. Multiple reviewers raise the question of novelty. The authors themselves agree that the paper is experimental in nature and thats how the merit of the paper should be determined.

One of the reviewers in the rebuttal mentions the following:
"The core contributions of a paper cannot be solely determined by a series of experimental results. More profound insights regarding these results should be carefully discussed, and what we could learn from those failure cases. Additionally, further experiments (more baselines and evaluate them on different datasets) should be conducted if they constitute the majority of the contributions."

I agree with reviewer - if this paper is experimental in nature than the experiments should at least provide insights and provide deep comparisons with state-of-the-art.

It is for this reason combined with in-general lack of strong support from any reviewer, that I recommend rejecting the manuscript.

**Justification For Why Not Higher Score:**

Multiple reviewers consider the paper below the threshold. The only positive reviewer didn't champion or participate in discussion.

**Justification For Why Not Lower Score:**

N/A

---

### Decision · Program_Chairs · 2024-01-16

Reject